# Blood Cultures and Appropriate Antimicrobial Administration after Achieving Sustained Return of Spontaneous Circulation in Adults with Nontraumatic Out-of-Hospital Cardiac Arrest

**DOI:** 10.3390/antibiotics10070876

**Published:** 2021-07-19

**Authors:** Chih-Hao Lin, Po-Lin Chen, Yi-Tzu Huang, Ching-Yu Ho, Chih-Chia Hsieh, William Yu Chung Wang, Ching-Chi Lee, Wen-Chien Ko

**Affiliations:** 1Department of Emergency Medicine, National Cheng Kung University Hospital, College of Medicine, National Cheng Kung University, Tainan 704, Taiwan; emergency.lin@gmail.com (C.-H.L.); hsiehchihchia@gmail.com (C.-C.H.); 2Department of Medicine, College of Medicine, National Cheng Kung University, Tainan 701, Taiwan; 3Department of Internal Medicine, National Cheng Kung University Hospital, College of Medicine, National Cheng Kung University, Tainan 704, Taiwan; cplin@mail.ncku.edu.tw; 4Department of Laboratory Medicine, Kaohsiung Medical University Hospital, Kaohsiung 807, Taiwan; eizi0384@yahoo.com.tw; 5Department of Adult Critical Care Medicine, Tainan Sin-Lau Hospital, Tainan 701, Taiwan; freebrid87@gmail.com; 6Department of Nursing, National Tainan Junior College of Nursing, Tainan 700, Taiwan; 7Department of Management Systems, University of Waikato, Hamilton 3240, New Zealand; william.wang@waikato.ac.nz; 8Clinical Medicine Research Center, National Cheng Kung University Hospital, College of Medicine, National Cheng Kung University, Tainan 704, Taiwan

**Keywords:** bacteremia, out-of-hospital cardiac arrest, return of spontaneous circulation, empirical antibiotic, mortality

## Abstract

We aimed to determine the incidence of bacteremia and prognostic effects of prompt administration of appropriate antimicrobial therapy (AAT) on nontraumatic out-of-hospital cardiac arrest (OHCA) patients achieving a sustained return of spontaneous circulation (sROSC), compared with non-OHCA patients. In the multicenter case-control study, nontraumatic OHCA adults with bacteremia episodes after achieving sROSC were defined as case patients, and non-OHCA patients with community-onset bacteremia in the emergency department were regarded as control patients. Initially, case patients had a higher bacteremia incidence than non-OHCA visits (231/2171, 10.6% vs. 10,430/314,620, 3.3%; *p* < 0.001). Compared with the matched control (2288) patients, case (231) patients experienced more bacteremic episodes due to low respiratory tract infections, fewer urosepsis events, fewer *Escherichia coli* bacteremia, and more streptococcal and anaerobes bacteremia. Antimicrobial-resistant organisms, such as methicillin-resistant *Staphylococcus aureus* and extended-spectrum beta-lactamase-producing *Enterobacteriaceae*, were frequently evident in case patients. Notably, each hour delay in AAT administration was associated with an average increase of 10.6% in crude 30-day mortality rates in case patients, 0.7% in critically ill control patients, and 0.3% in less critically ill control patients. Conclusively, the incidence and characteristics of bacteremia differed between the nontraumatic OHCA and non-OHCA patients. The incorporation of blood culture samplings and rapid AAT administration as first-aids is essential for nontraumatic OHCA patients after achieving sROSC.

## 1. Introduction 

Bloodstream infections are associated with substantial morbidity and mortality that cause a significant burden of healthcare costs [1]. Community-onset bacteremia was reported to have an annual incidence of 0.14−0.82% [2,3] for the general population in a community, and a short-term fatality rate of up to 41.5% [4]. In addition to hemodynamic support, prompt administration of appropriate antimicrobial therapy (AAT) has been evidenced to provide survival benefits for patients with bacteremia, particularly those who are critically ill on initial presentation [4,5,6,7].

While the incidence and survival rate of out-of-hospital cardiac arrest (OHCA) varies in different areas, it vastly leads to a universally miserable outcome [8]. Although many studies have reported infection or sepsis complication secondary to therapeutic hypothermia, and its episode, in numerous days after return of spontaneous circulation (ROSC) in OHCA patients [9,10,11], little is known about the incidence and characteristics of bloodstream infections in OHCA patients achieving ROSC [12]. Additionally, in the contemporary resuscitation guideline issued by the American Heart Association (AHA), sampling of blood cultures is not routinely incorporated in the “chain of survival” [13] for OHCA patients. Therefore, the emergency department (ED) physicians do not include blood cultures to be a routine test after achieving ROSC and do not administer antimicrobial therapy as the first aid for patients initially experiencing OHCA. We hypothesized that if OHCA presentation is accompanied by a high incidence of bacteremia and prompt AAT administration has a beneficial effect on the short-term prognoses of OHCA patients with bacteremia, then rapid AAT administration would thus be crucial for surviving nontraumatic OHCA patients after achieving ROSC. This study aimed to investigate the characteristics and incidence of bacteremia in nontraumatic OHCA adults compared with non-OHCA patients, and also to elucidate the prognostic effect of prompt AAT administration on those achieving sustained ROSC (sROSC).

## 2. Materials and Methods 

### 2.1. Study Design and Setting 

This case-control multicenter study was conducted with datasets recorded from January 2008 to December 2020 at EDs of three hospitals, with 1200, 460, and 380 beds, in southern Taiwan. The case patients were nontraumatic OHCA adults with bacteremia episodes after achieving sROSC. The control patients were non-OHCA patients with community-onset bacteremia recognized in the ED. All resuscitation in the targeted hospitals was operated by board-certified ED physicians and nurses following the contemporary guidelines of advanced cardiac life support. The study was approved by the institutional review boards of the aforementioned medical institutes without the requirement for informed consent, and was reported in the format recommended in the STROBE criteria (Strengthening the Reporting of Observational Studies in Epidemiology).

### 2.2. Selection of Participants 

Medical records of all ED visits during the study period were retrospectively reviewed in a computer database for data gathering. Of adult patients upon OHCA presentation, those with nontraumatic OHCA who had blood cultures sampled within one day after achieving sROSC were included as case patients, after exclusions of those with contaminant sampling. 

During the study period, non-OHCA patients with bacterial growth of blood cultures sampled in the ED were assigned to the group of community-onset bacteremia, after the exclusions of those with contaminant sampling, with hospital-onset bacteremia, who had been transferred from other hospitals, and diagnosed with bacteremia before ED arrival. Moreover, ten patients who had community-onset bacteremia temporally near to the time of ED arrival were matched as the non-OHCA comparators (i.e., the control patients). Finally, to study the impacts of delayed AAT administration on short-term prognoses, for both the case and control patients, those with an uncertain mortality or incomplete clinical information within 30 days of bacteremia onset were excluded. 

### 2.3. Measurements and Outcomes

For case and control patients, clinical data were collected jointly by one board-certified ED physician and another infectious-disease clinician who had received appropriate training in medical chart reviews, and both were blinded to the aim and hypotheses of the present study; furthermore, recording discrepancy was solved by discussion between the authors. Using a predetermined form, the extracted data consisted of demographic information, vital signs, comorbidities, comorbidity severity (McCabe–Johnson classification), laboratory data, and bacteremia severity (a Pitt bacteremia score) at ED arrival, bacteremia sources, durations and types of antimicrobial therapy, and causative microorganisms. Moreover, frequencies of patients’ ED visits within six months before the bacteremia episode, subsequent discharge, or hospitalization through the ED, and short-term outcomes were obtained. Focusing on case patients, this study captured more information about the timing of sROSC initiation and blood culture sampling. The primary outcome was crude 30-day mortality after bacteremia onset (i.e., ED arrival).

### 2.4. Sampling of Blood Cultures and Microbiological Methods 

During the study period, blood sampling was performed by nurses immediately upon ED arrival, and two culture sets were routinely taken from different central or peripheral vessels with at least a 30-min interval between the two samplings. One culture set routinely consisted of each aerobic and anaerobic culture, with approximately 5–8 mL blood in one culture bottle. After sampling, four bottles of blood cultures were incubated in a BACTEC 9240 instrument (Becton Dickinson Diagnostic Systems, Sparks, MD, USA) for 5 days at 35 °C. The causative microorganisms were identified by a semiautomated system (Vitek 2 system, bioMe’rieux, Durham, NC, USA), and were prospectively stored for further susceptibility testing (the disk diffusion method for aerobes and the agar dilution method for anaerobes), to identify antimicrobial-resistant organisms (AROs) and to determine the AAT timing for each eligible patient, provided that the susceptibility to antimicrobials administered by ED clinicians was not provided by the study hospital.

To further investigate the ARO incidence in the community, the susceptibility to ampicillin was tested for enterococci, cefoxitin for *Staphylococcus aureus* (to reflect methicillin susceptibility), penicillin for streptococci, and ampicillin/sulbactam for anaerobes. Focusing on *Escherichia coli*, *Klebsiella* species, and *Proteus mirabilis* (EKP), susceptibilities to cefepime and levofloxacin were tested. Their production of extended-spectrum beta-lactamases (ESBLs) was further determined by the phenotypic confirmatory test of cephalosporin–clavulanate combination disks. All susceptibilities were interpreted by the contemporary recommendation issued by the Clinical and Laboratory Standards Institute (CLSI) [14].

### 2.5. Definitions

Cardiac arrest was diagnosed on the basis of the latest AHA definition [13]. The term ‘bacteremia′ refers to the bacterial growth of blood cultures obtained from central or peripheral venipuncture, after exclusion of contaminant sampling. Blood cultures with the growth of coagulase-negative staphylococci, *Bacillus *spp., *Micrococcus* spp., *Propionibacterium* spp., or Gram-positive bacilli, were considered as contaminant samplings based on the previously established criteria [15]. Community-onset bacteremia indicates that the place of bacteremia onset is the community [5,7]. Polymicrobial bacteremia was defined as the isolation of more than one microbial species from a single bacteremic episode, whereas the other was monomicrobial bacteremia.

As suggested by the extant literature [5,7], antimicrobial therapy was appropriate if the following two criteria were fulfilled: (i) the route and dosage of antimicrobials were administered as recommended in the *2021 Sanford Guide* [16]; (ii) antimicrobials administered were in vitro active against all causative microorganisms of bacteremia, based on the CLSI breakpoints issued in 2021 [14]. For the case patients, the time-to-appropriate antibiotic (TtAa) was defined as the time gap between the first dose of AAT administration and the initiation of sROSC; for the control patients, the gap between the first dose of AAT administration and ED arrival was defined as the TtAa [5,7].

Comorbidities were defined as described previously [17], and the comorbidity severity was graded according to the classification system proposed by McCabe and Johnson [18]. Bacteremia severity was graded in accordance with a Pitt bacteremia score (PBS) using a previously validated scoring system immediately upon ED arrival [5,7]. Patients having a PBS of ≥4 points were categorized as having critical illness, whereas those with a PBS of <4 points were classified as having less critical illness. Bacteremia sources were clinically identified according to one of the following criteria: the presence of an active infection site coincident with bloodstream infections; the isolation of a microorganism from other clinical specimens before or on the same date as that of bacteremia onset [19]. For complicated bacteremia, removal of infected hardware, drainage of infected fluid collections, or resolution of obstruction for biliary or urinary sources was referred to as appropriate control of bacteremia source [20]. Crude mortality was equated with death from all causes [7].

### 2.6. Statistical Analyses

The Statistical Package for the Social Science for Windows (Version 23.0; Chicago, IL, USA) was used for statistical analyses of this study. Categorical variables were compared using the Fisher exact or Pearson *Chi*-square test. Continuous variables were presented as medians (interquartile ranges, IQRs) and were compared using an independent *t*-test or Mann–Whitney U test.

To investigate the independent effect of the TtAa (each hour) on 30-day mortality, respectively, in case, critically ill (a PBS ≥ 4) control, and less critically ill (a PBS < 4) control patients, the variables of 30-day crude mortality, recognized through the univariate analysis with a *p* value of <0.05, (or *p* < 0.1 for a small patient population), and the TtAa were together included in a stepwise and backward multivariable logistic regression model. A TtAa-related trend in the 30-day crude mortality rate was assessed by the Spearman’s correlation. A E-value was calculated to examine the potential effect of unmeasured confounders inside our study [21]. A two-sided *p* value of ≤0.05 was considered statistically significant.

## 3. Results 

### 3.1. Characteristics of Study Subjects 

During the study period, OHCA patients accounted for 0.68% (8656) of the total 1,272,388 ED visits. Patients achieving sROSC constituted 28.4% (2203 patients) of those with nontraumatic OHCA. The contamination rates of blood cultures (7.1% vs. 1.8%, *p* < 0.001) and true bacteremia incidence (10.6% vs. 3.3%, *p* < 0.001) in nontraumatic OHCA patients with sROSC were significantly higher than those in non-OHCA patients (Figure 1). Based on the inclusion and exclusion criteria, the study cohort of community-onset bacteremia consisted of 231 case patients and 2288 matched control patients (Figure 1).

Of 231 case patients, their median (interquartile range (IQR)) age was 73 (57–82) years old, and 121 (52.4%) were male. The length (median (IQR)) of the time gap from ED arrival to the initiation of sROSC, namely the time-to-sROSC, was 11 (5–19) mins, and the gap between the timing of the initial sROSC and the sampling of blood cultures was 0.6 (0.4–2.7) hours. After sROSC, 141 (61.0%) were admitted to intensive care units (ICUs), 81 (35.1%) died at the ED, and 9 (3.9%) were admitted to general wards. The median (IQR) length of ED stay and hospitalization was 7.5 (2.6–21.9) hours and 2.1 (0.8–9.0) days.

Among 2288 control patients, the elderly accounted for 61.1% (1399 patients), and 50.8% (1162) were male. The median (IQR) length of ED stay and was 15.5 (5.7–26.0) hours and 10.0 (5.9–18.1) days. Most patients (1918, 83.8%) were admitted to general wards, 205 (9.0%) to ICUs, 47 (2.1%) died at the ED, and 118 (5.2%) were directly discharged from the ED and followed up in the out-patient clinic. 

### 3.2. Clinical Characteristics and Outcomes of Case and Control Patients

The comparisons of clinical variables between case and control patients, in terms of patient demographics, laboratory data and bacteremia severity at ED arrival, types and severity of comorbidities, the timing of AAT administration, and patient outcomes, were examined by the univariate analyses (Table 1). The case patients were more likely to be nursing-home residents, with a bed-ridden status, as well as being more likely to have critical illness (a PBS ≥ 4), polymicrobial bacteremia, bacteremia caused by low respiratory tract infections, high leucocytes or c-reactive protein (CRP), and comorbidities of neurological or psychological diseases. However, bacteremia due to urinary tract, skin and soft-tissue, intra-abdominal, or biliary tract infections, and comorbid liver cirrhosis were less frequently disclosed in the case patients. Notably, a longer TtAa and more frequencies of prior ED visits were observed in the case patients, and the crude 3-day, 15-day, or 30-day mortality rate in the case patients was higher than that in the control patients.

### 3.3. Microorganisms and Susceptibilities in Case and Control Patients

Because of 43 and 219 polymicrobial bacteremia in the case and control patients, 316 and 2568 causative microorganisms were, respectively, collected (Figure 2A). The five most commonly identified genera/species in the case patients were *Streptococcus* species (51 isolates, 16.1%), *Klebsiella* species (50, 15.8%), *Escherichia coli* (48, 15.2%), *Staphylococcus aureus* (35, 11.1%), and anaerobes (24, 7.6%). Notably, *Streptococcus pneumoniae* and *Streptococcus agalactiae* accounted for 62.7% (32 isolates) and 21.6% (11) of all streptococci, respectively. Otherwise, the leading five microorganisms in the control patients were the same as those in the case patients, but less streptococci, less anaerobes, and more *E. coli* were observed in the control patients, compared to the case patients (Figure 2A). Additionally, the three major anaerobe genera were *Fusobacterium* (11 isolates, 45.8%), *Peptostreptococcus* (5, 20.8%), and *Prevotella* (4, 16.7%) in the case patients, whereas the most commonly identified anaerobes in the control patients were *Bacteroides* (74 isolates, 67.2%), *Fusobacterium* (12, 10.9%), and *Peptostreptococcus* (8, 7.2%). Notably, the distribution diversity of anaerobe species between the case and control patients was disclosed.

For the common AROs in the community, more AROs, including MRSA, penicillin-resistant streptococci, ampicillin/sulbactam-resistant anaerobes, and ESBL-producing or levofloxacin-resistant EKP, were noted in the case patients, compared to the control patients (Figure 2B).

### 3.4. Impacts of Delayed AAT on Mortality of Patients with Varied Bacteremia Severity

For the case patients, clinical predictors of crude 30-day mortality in the univariate analyses, in terms of fatal comorbidities (McCabe–Johnson classification) and comorbidities of psychological and cardiovascular diseases, were identified (Table 2). The length (measured by hour) of the TtAa remained as the crucial determinant (adjusted odds ratio (AOR), 1.106; *p =* 0.005) of 30-day mortality after adjusting two independent predictors, in terms of fatal comorbidities and the time-to-sROSC of >11 min, recognized by the multivariate regression model. 

For the critically ill control patients, the TtAa remained as an independent determinant of 30-day mortality (AOR, 1.007; *p* < 0.001), after adjustment of three independent predictors (i.e., fatal comorbidities, and bacteremia due to urinary or low respiratory tract infections) recognized by the multivariate regression model (Table 2). Furthermore, focusing on the less critically ill control patients (Table 2), the prognostic effect of the TtAa remained significant (AOR, 1.003; *p <* 0.001), after adjusting independent predictors of 30-day mortality, in terms of nursing-home residents, polymicrobial bacteremia, bacteremia due to urinary or low respiratory tract infections, fatal comorbidities, and comorbidities of malignancies or chronic obstructive pulmonary diseases.

In sum, each hour of delayed AAT administration resulted in an average increase of 10.6% (AOR, 1.106; *p* = 0.005), 0.7% (AOR, 1.007; *p* < 0.001), and 0.3% (AOR, 1.003; *p* = 0.004) of 30-day crude mortality rates in the case, critically ill control, and less critically ill control patients, respectively. Notably, the greatest impact of the delayed TtAa (i.e., the highest AOR value) on 30-day mortality among three patient groups was recognized in the case patients. In further analysis, a positive trend of the TtAa in the 30-day mortality rate among the case (*γ* = 0.986, *p* < 0.01), critically ill control (*γ* = 1.000, *p* < 0.01), and less critically ill control patients (*γ* = 1.000, *p* = 0.01) was evident in Figure 3. 

## 4. Discussion

The present study demonstrated a higher contamination rate of blood cultures and concurrent bacteremia incidence in nontraumatic OHCA patients after achieving sROSC, compared with those who were non-OHCA at the ED. Notably, significant differences were identified in bacteremia sources, causative microorganisms, and antimicrobial susceptibilities between the two patient groups with bacteremia. Compared with non-OHCA patients with community-onset bacteremia, the data from the OHCA patients exhibited a greater likelihood of having low respiratory tract infections and a lower likelihood of having urosepsis. Focusing on causative microorganisms, this phenomenon was compatible with more bacteremic events due to anaerobes and streptococci, but fewer events due to *E. coli* in OHCA patients herein. More importantly, frequent bacteremia episodes caused by AROs and the survival benefit of prompt AAT administration in OHCA patients were first recognized herein. Accordingly, the clinical practice of blood culture samplings after achieving sROSC and the incorporation of appropriate antimicrobials, as empirical therapy, into the antibiotic stewardship program should be considered as the first aids for OHCA patients.

Consistent with previous reports [12], bacteremia episodes were frequent in OHCA presentations in our cohort. Although bacterial translocation raised from intestinal ischemia during the peri-arrest period might result in secondary bloodstream infections [22], there are three convincing reasons to support the episodes of community-onset bacteremia in nontraumatic OHCA patients herein, and to not view them as the consequence of resuscitation. First, the period between the sampling of blood cultures and ED arrival was noted to be extremely short in our case patients. Second, serum CRP levels were upregulated within 4 to 6 h after sepsis or infection stimulation, doubled after every 8 h, and peaked after 35 to 60 h in patients experiencing bloodstream infections or sepsis [23]; therefore, the high serum CRP levels observed in the nontraumatic OHCA patients indicated that bacteremia occurred prior to resuscitation. Third, because the predominant bacteremia-causing microorganisms in nontraumatic OHCA patients were streptococci, particularly *S. pneumoniae*, rather than Enterobacteriaceae, the possibility of bacterial translocation from intestinal ischemia might be negligible.

To our knowledge, a delay in AAT administration increases the risk of unfavorable prognoses in adults with community-onset bacteremia; in addition, for more severe bacteremia episodes, faster AAT administration is recommended [5,7]. Therefore, compared to non-OHCA patients with bacteremia, the greater impact of delayed AAT administration on prognoses of patients with bacteremia, initially presenting an extremely critical situation with cardiac arrest, was reasonably disclosed in our cohort. Although optimization in the “chain of survival” has been evidenced to improve survival and neurological outcomes of OHCA patients [24,25], the appropriateness of empirical antimicrobial therapy has not been integrated into the strategy. On the basis of our findings, rapid AAT administration might be incorporated into the “chain of survival” to achieve favorable outcomes in nontraumatic OHCA patients who have risk factors for bloodstream infections and have achieved sROSC.

Contaminated blood cultures adversely affect health care and medical expenditures, causing unnecessary hospitalizations, antimicrobial administration, and microbiological studies [26]. Furthermore, ED overcrowding is a growing problem worldwide and is associated with blood culture contamination [27], delayed administration of appropriate antibiotics [28,29], and unfavorable patient outcomes [29]. Patients with cardiac arrest are emergently resuscitated and comprehensively evaluated, which substantially increases the utilization of emergency care resources and the degree of ED overcrowding. Therefore, a high contamination rate of blood culture samples in patients experiencing OHCA is not unexpected.

Based on this definition [13], OHCA occurs within the community. Although the blood culture was sampled after sROSC achievement, bacteremia detected in patients with OHCA after resuscitation was reasonably regarded as community-onset; this is because the period between blood culture sampling and ED arrival was extremely short and because few reports have indicated an association between nosocomial bacteremia episodes and resuscitation. Therefore, non-OHCA patients diagnosed as having community-onset bacteremia in the ED were referred to as reasonable controls for the present study. In our study designs, the case patients were selected at a ratio of 10:1, according to the arrival timing of each case patient. Such ED encounters were sufficiently representative of the comparators of community-onset bacteremia for the following reasons. First, the main causative microorganisms, namely *E. coli*, *S. aureus*, *Streptococcus*species, and *Klebsiella *species, as well as common bacteremia sources, such as urosepsis, intra-abdominal infections, and soft-tissue infections, in the matched control patients, were consistent with those in previously established cohorts with community-onset bacteremia [2,5,7,30]. Second, the proportion of critically ill patients at bacteremia onset (approximately 20%) and the crude mortality rate (15%) in the present cohort were also consistent with the corresponding data reported in other studies [2,5,7,30].

Of AROs in the community, ESBL-producers have well known adverse effects on the prognoses of bacteremic patients in the literature [31]. Clinical predictors of ESBL-producers in community-onset bacteremia, including healthcare facility residence, urinary catheter use, previous antimicrobial therapy, frequent ED visits [32], and previous hospitalization, have been reported [31]. Studies assessing the frequency of ED visits have suggested that it may be correlated with increased healthcare use, serious ill health, and socioeconomic distress [33,34]. Accordingly, a strong association of antimicrobial-resistant microorganisms and bacteremic patients experiencing OHCA reasonably results from frequent utilization of medical care.

Our findings might be interpreted with caution as a result of several limitations that are inherent to the study design. First, the study is majorly limited due to its retrospective and observational nature. To improve accuracy and minimize inconsistencies in medical chart reviews, in line with previous suggestions [35], data were captured by the appropriately trained physicians and these chart abstractors were blind to the aim and hypotheses in our study. Second, the data were collected from three hospitals located in southern Taiwan, and thus, there might be a limitation to the validation of our principal implications on other communities, despite the large sample frame of approximately 2000 adults that are representative of community-onset bacteremia in a longitudinal context. Third, a low E-value was obtained for the prognostic effects of delayed EAT administration on various patients (i.e., 1.28 in the case patients, 1.06 in the critically ill case patients, and 1.05 in less critically ill case patients); thus, unmeasured confounders in our cohort should be trivial. Fourth, to avoid the selection bias caused by the misclassification of contaminated blood sampling, we followed the previously established criteria [15] that has been adapted in numerous studies to identify true bacteremia episodes. Finally, OHCA or non-OHCA patients who met the exclusion criteria constituted only a small portion of the overall population. Selection bias may have exerted a trivial influence on our results. Of importance, the prognostic advantage of prompt AAT administration after achieving sROSC in nontraumatic OHCA patients with bacteremia was first emphasized here. As for future research, prospective studies from other regions to evaluate the incidence and significance of bacteremia, as well as to identify the bacteremia predictor in OHCA patients, would contribute to the improved quality of ED management.

## 5. Conclusions

Because of the high incidence of bacteremia in nontraumatic OHCA patients after achieving sROSC, the sampling of blood cultures might be performed after successful resuscitation for those having risk factors of bloodstream infections. To improve care quality, the incorporation of prompt AAT administration, after achieving sROSC, into the strategy of “chain of survival” is warranted for nontraumatic OHCA patients. However, further study is warranted to identify the cause–effect relationship between cardiac arrest and bacteremia episodes and to determine whether bacteremia is the immediate byproduct of cardiac arrest or is the leading contributing factor to unidentified severe sepsis and septic shock, which results in the sudden onset of cardiac arrest.

## Figures and Tables

**Figure 1 antibiotics-10-00876-f001:**
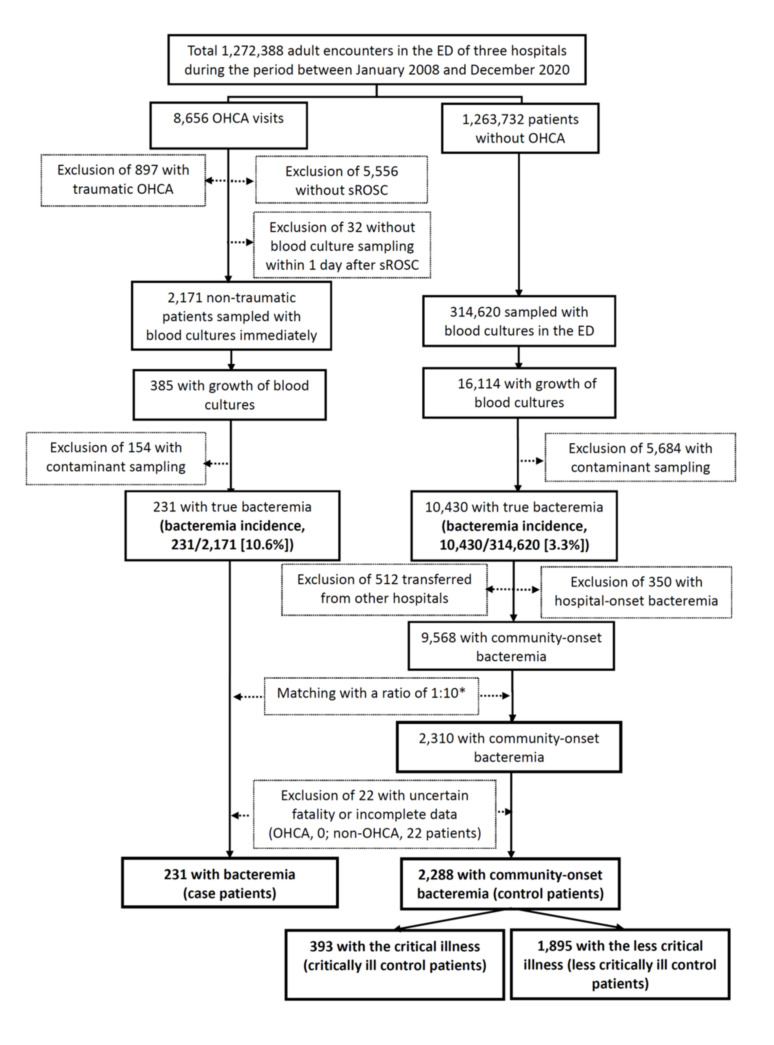
Flowchart of patient selections. BC = blood culture; ED = emergency department; OHCA= out-of-hospital cardiac arrest; sROSC = sustained return of spontaneous circulation. * Ten non-OHCA patients temporally near ED arrival of each OHCA visit were matched.

**Figure 2 antibiotics-10-00876-f002:**
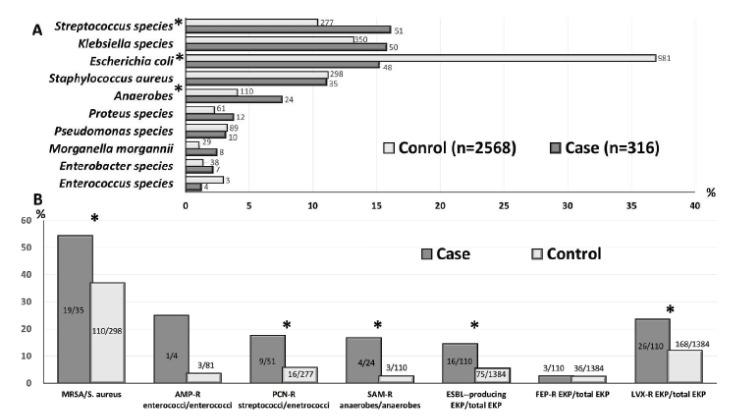
Distribution of the leading ten microorganisms (**A**) ^#^ and antimicrobial-resistant microorganisms (**B**) ^§^ between the case and control patients. AMP = ampicillin; EKP = *Escherichia coli*, *Klebsiella* species, and *Proteus mirabilis*; ESBL = extended-spectrum beta-lactamase; LVX = levofloxacin; MRSA = methicillin-resistant *Staphylococcus aureus*; PCN = penicillin; R = resistant; SAM = ampicillin/sulbactam. * indicates a significant difference (i.e., *p* < 0.05) between the case and control patients. ^#^ Numbers in the bar indicate the isolate numbers. ^§^ The denominators and numerators, respectively, indicate the isolate numbers of all microorganisms and antimicrobial-resistant pathogens.

**Figure 3 antibiotics-10-00876-f003:**
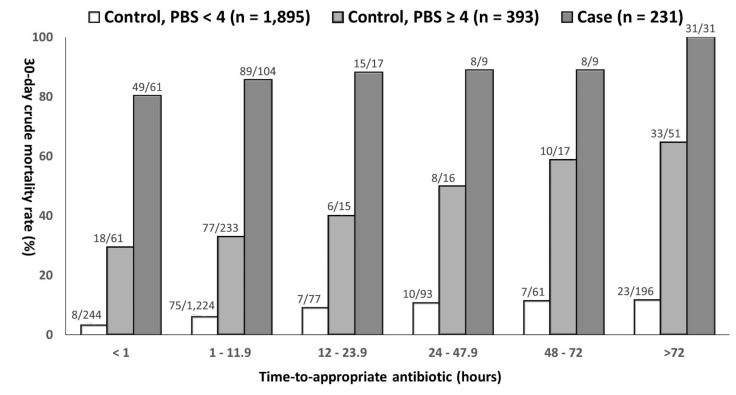
A positive trend of the time-to-appropriate antibiotic in the 30-day crude mortality rate in bacteremic patients, categorized into the case, critically ill (a Pitt bacteremia score ≥ 4) control, and less critical ill (a Pitt bacteremia score < 4) control patients. The denominators and numerators, respectively, indicate case numbers of crude 30-day mortality and all patients in varied categories of the time-to-appropriate antibiotic.

**Table 1 antibiotics-10-00876-t001:** Clinical manifestations and outcomes of community-onset bacteremia in case and control patients *.

Variables	Patient Numbers (%)		*p* Value
	Case *n* = 231	Control *n* = 2288	
Patient demographics			
Gender, male	121 (52.4)	1162 (50.8)	0.64
The elderly, ≥65 years	153 (66.2)	1399 (61.1)	0.13
**Nursing-home residents**	**34 (14.7)**	**127 (5.6)**	**<0.001**
**Bed-ridden status**	**82 (35.5)**	**289 (12.6)**	**<0.001**
**ED visits within prior 6 months, median (IQR)**	**2 (1–3)**	**0 (0–1)**	**<0.001**
**Polymicrobial bacteremia**	**43 (18.6)**	**219 (9.6)**	**<0.001**
**Pitt bacteremia score ≥4 at ED arrival**	**231 (100)**	**393 (17.2)**	**<0.001**
Major sources of bacteremia			
**Low respiratory tract infections**	**136 (58.9)**	**302 (13.2)**	**<0.001**
**Urinary tract infections**	**17 (7.4)**	**765 (33.4)**	**<0.001**
**Skin and soft-tissue infections**	**14 (6.1)**	**233 (10.2)**	**0.045**
**Intra-abdominal infections**	**10 (4.3)**	**293 (12.8)**	**<0.001**
**Biliary tract infections**	**4 (1.7)**	**207 (9.0)**	**<0.001**
Ultimately or rapidly fatal comorbidities (McCabe–Johnson classification)	59 (25.5)	545 (23.8)	0.56
Major comorbidities			
Cardiovascular diseases	125 (54.1)	1211 (52.9)	0.73
**Neurological diseases**	**96 (41.6)**	**545 (23.8)**	**<0.001**
Diabetes mellitus	95 (41.1)	849 (37.1)	0.23
Malignancies	70 (30.3)	647 (28.3)	0.52
Chronic kidney diseases	45 (19.5)	417 (18.2)	0.64
Urological diseases	19 (8.2)	154 (6.7)	0.39
Chronic obstructive pulmonary diseases	18 (7.8)	113 (4.9)	0.06
**Liver cirrhosis**	**17 (7.4)**	**293 (12.8)**	**0.02**
**Psychological diseases**	**14 (5.1)**	**33 (1.4)**	**<0.001**
**Time-to-appropriate antibiotic, hours, median (IQR)**	**10 (0.9–20.0)**	**2.0 (1.1–8.0)**	**<0.001**
Laboratory data at EDs, median (IQR)			
**Leukocyte (1000/mm^3^)**	**13.5 (7.1–20.6)**	**11.5 (7.5–16.3)**	**<0.001**
**C-reactive protein (mg/L), *n* = 2734**	**121.3 (41.7–214.3)**	**69.9 (28.3–186.3)**	**<0.001**
Crude mortality rates			
**3-day**	**142 (61.5)**	**99 (4.3)**	**<0.001**
**15-day**	**177 (76.6)**	**195 (8.5)**	**<0.001**
**30-day**	**200 (86.6)**	**282 (12.3)**	**<0.001**

ED = emergency department; IQR = interquartile range. * Boldface indicates statistical significance with a *p* value of ≤ 0.01.

**Table 2 antibiotics-10-00876-t002:** Impacts of the time-to-appropriate antibiotic on 30-day prognoses in patients with bacteremia, categorized as case, critically ill (Pitt bacteremia score ≥ 4) control, and less critically ill (Pitt bacteremia score < 4) control patients.

Clinical Variables	Patient Number (%)		Univariate Analysis		Multivariate Analysis	
	Death	Survival	OR (95% CI)	*p*-Value	AOR (95% CI)	*p* Value
Case patients (*n* = 231)	*n* = 200	*n* = 31				
**Time-to-appropriate antibiotic (hour) ***	–	–	–	–	**1.106 (1.031–1.187)**	**0.005**
**Time-to-sROSC ** > 11 min**	**114 (57.0)**	**3 (9.7)**	**12.4 (3.6–42.0)**	**<0.001**	**19.00 (5.32–67.83)**	**<0.001**
**Ultimately or rapidly fatal comorbidities (McCabe–Johnson classification)**	**58 (29.0)**	**1 (3.2)**	**12.25 (1.63–91.97)**	**0.002**	**8.82 (1.09–72.28)**	**0.04**
Comorbidities						
Cardiovascular diseases	103 (51.5)	22 (71.0)	0.43 (0.19–0.99)	0.04	NS	NS
Malignancies	65 (32.5)	5 (16.1)	2.50 (0.92–6.82)	0.07	NS	NS
Psychological diseases	9 (4.5)	5 (16.1)	0.25 (0.08–0.79)	0.01	NS	NS
Critically ill control patients (*n* = 393)	*n* = 152	*n* = 241				
**Time-to-appropriate antibiotic (hour) ***	–	–	–	–	**1.007 (1.004–1.010)**	**<0.001**
Gender, male	99 (65.1)	130 (53.9)	1.60 (1.05–2.43)	0.03	NS	NS
Inadequate source control	8 (5.3)	6 (2.5)	2.18 (0.74–6.40)	0.15	2.72 (0.83–8.87)	0.10
Bacteremia sources						
**Urinary tract infections**	**17 (11.2)**	**79 (32.8)**	**0.26 (0.15–0.46)**	**<0.001**	**0.44 (0.23–0.83)**	**0.01**
**Low respiratory tract infections**	**62 (40.8)**	**62 (25.7)**	**1.99 (1.29–3.06)**	**0.002**	**1.74 (1.05–2.89)**	**0.03**
**Ultimately or rapidly fatal comorbidities (McCabe–Johnson classification)**	**70 (46.1)**	**58 (24.1)**	**2.69 (1.74–4.18)**	**<0.001**	**2.42 (1.51–3.89)**	**<0.001**
Comorbidities						
Malignancies	70 (46.1)	66 (27.4)	2.26 (1.48–3.47)	<0.001	NS	NS
Diabetes mellitus	52 (34.2)	105 (43.6)	0.67 (0.441–1.03)	0.06	0.63 (0.39–1.006)	0.05
Neurological diseases	48 (31.6)	98 (40.7)	0.67 (0.44–1.03)	0.07	NS	NS
Liver cirrhosis	28 (17.1)	19 (7.9)	2.41 (1.26–4.53)	0.005	NS	NS
Urological diseases	8 (5.3)	32 (13.3)	0.36 (0.16–0.81)	0.01	NS	NS
COPD	7 (4.6)	25 (10.4)	0.42 (0.18–0.99)	0.04	0.40 (0.15–1.06)	0.07
Less critically ill control patients (*n* = 1895)	*n* = 130	*n* = 1765				
**Time-to-appropriate antibiotic (hour) ***	–	–	–	–	**1.003 (1.001 −1.005)**	**0.004**
Inadequate source control	6 (4.6)	45 (2.5)	1.85 (0.77–4.42)	0.16	2.35 (0.92 − 6.05)	0.08
Gender, male	75 (57.7)	858 (48.6)	1.44 (1.01–2.07)	0.046	NS	NS
**Nursing-home residents**	**14 (10.8)**	**58 (3.3)**	**3.55 (1.92–6.56)**	**<0.001**	**4.27 (2.14–8.54)**	**<0.001**
**Polymicrobial bacteremia**	**20 (15.4)**	**136 (7.7)**	**2.18 (1.31–3.62)**	**0.002**	**2.04 (1.17–3.54)**	**0.01**
Bacteremia sources						
**Low respiratory tract infections**	**31 (23.8)**	**147 (8.3)**	**3.45 (2.23–5.34)**	**<0.001**	**2.07 (1.26–3.43)**	**0.04**
Intraabdominal infections	26 (20.0)	227 (12.9)	1.69 (1.08–2.66)	0.02	NS	NS
Urinary tract infections	21 (16.2)	648 (36.7)	0.33 (0.21–0.54)	<0.001	0.50 (0.29–0.85)	0.01
**Ultimately or rapidly fatal comorbidities (McCabe–Johnson classification)**	**76 (58.5)**	**341 (19.3)**	**5.88 (4.07–8.49)**	**<0.001**	**3.65 (2.33–5.72)**	**<0.001**
Comorbidities						
**Malignancies**	**74 (56.9)**	**437 (24.8)**	**4.02 (2.79–5.78)**	**<0.001**	**1.98 (1.26–3.13)**	**0.003**
Liver cirrhosis	35 (26.9)	213 (12.1)	2.68 (1.78–4.06)	<0.001	NS	NS
**COPD**	**13 (10.0)**	**68 (3.9)**	**2.77 (1.49–5.17)**	**0.001**	**2.44 (1.20–4.94)**	**0.01**

AOR = adjusted odds ratio; CI = confidence interval; COPD = Chronic obstructive pulmonary diseases; NS = not significant (after processing the backward multivariate regression); OR = odds ratio. * The time-to-appropriate antibiotic, a continuous variable, was included in the multivariable logistic regression model; boldface indicates statistical significance with a *p* value of ≤ 0.01 under the multivariate regression model. ** The time-to-sROSC indicated the time gap from ED arrival to the initiation of sROSC.

## Data Availability

Data are available from the corresponding authors on reasonable request.

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
