# Peer review of "Blood Cultures and Appropriate Antimicrobial Administration after Achieving Sustained Return of Spontaneous Circulation in Adults with Nontraumatic Out-of-Hospital Cardiac Arrest"

_antibiotics, 2021, doi:10.3390/antibiotics10070876_

Round 1

Reviewer 1 Report

The article submitted for review is of great interest since it shows that there is a significant percentage of bacteremia in patients who suffer cardio-respiratory arrest. This fact has already been studied in previous studies. The strength of the article lies in the number of patients recruited. A weakness of the article could be that there is a percentage of contaminated cultures that could modify the results. Another weakness would be that it has only been tested in one health system. Possibly the results could be different in another country with a better or worse health system in which cardiac arrests were greater or lesser with an infectious cause. 
Undoubtedly, one of the future lines of study in the future would be to determine the clinical usefulness of early administration of broad-spectrum antibiotherapy in cardiac arrest. 

Author Response

The article submitted for review is of great interest since it shows that there is a significant percentage of bacteremia in patients who suffer cardio-respiratory arrest. This fact has already been studied in previous studies. The strength of the article lies in the number of patients recruited. A weakness of the article could be that there is a percentage of contaminated cultures that could modify the results. Another weakness would be that it has only been tested in one health system. Possibly the results could be different in another country with a better or worse health system in which cardiac arrests were greater or lesser with an infectious cause. Undoubtedly, one of the future lines of study in the future would be to determine the clinical usefulness of early administration of broad-spectrum antibiotic therapy in cardiac arrest. 

Response: Many thanks for your substantial opinions and suggestions. We agree your opinions detailing the two weakness in our study. To avoid the selection bias we had followed the previously established criteria (Line 162-164, Page 4 and Reference No. 15) to distinguish the contaminated sampling from true bacteremia. We have also addressed our concerning and efforts in the paragraph of limitation (Line 394-397, Page 9). Another weakness is the limited validation of our principal finding on other community. We had emphasized this issue in the in the paragraph of limitation (Line 387-391, Page 9).

Reviewer 2 Report

I evaluated the manuscript and I confirm that in my opinion is ready for publication. I don't have further comments to improve quality of this paper.

Author Response

I evaluated the manuscript and I confirm that in my opinion is ready for publication. I don't have further comments to improve quality of this paper.

Response: Many thanks for your review.

Reviewer 3 Report

In this manuscript the authors have investigated the incidence of bacteremia and the prognostic effects of prompt administration of antimicrobial therapy (AAT) on nontraumatic out-of-hospital cardiac arrest (OHCA) patients achieving a sustained return of spontaneous circulation (sROSC), compared with non-OHCA patients. The incidence and characteristics of bacteremia were different between the nontraumatic OHCA and non-OHCA patients. It was concluded that performing blood culture sampling and rapid AAT administration as first-aids was essential for nontraumatic OHCA patients following sROSC.

I suggest that this manuscript would be suitable for publication in Antibiotics if subjected to some minor grammatical and typographical changes as follows:

Line 61.  Change ‘the’ to ‘a’

Line 64.  Change ‘as’ to ‘on’

Line 71.  Change ‘regarding’ to ‘about’ and ‘characteristic’ to ‘characteristics’

Line 88.  Change ‘The case’ to ‘This case’, ‘has been’ to ‘was’ and ‘with the datasets’ to ‘with datasets’

Line 90.  Remove ‘, respectively,’ and change ‘The case patient was the’ to ‘Case patients were’

Line 91.  Change ‘adult’ to ‘adults’

Lines 91-92.  Change ‘The control patient was the non-OHCA patient’ to ‘Control patients were non-OHCA patients’

Line 101.  Change ‘record’ to ‘records’

Line 114.  Remove ‘the’

Line 120.  Change ‘well’ to ‘appropriate’

Line 121.  Change ‘and the recording’ to ‘and recording’

Lines 121-122.  Change ‘by the discussion’ to ‘by discussion’

Line 123.  Change ‘consist’ to ‘consisted’

Line 135.  Change ‘done’ to ‘taken’

Line 195.  Change ‘for the statistical analyses for this study’ to ‘for statistical analysis of this study’

Line 212.  Remove ‘who’

Line 239.  Change ‘was’ to ‘were’

Line 240.  Remove ‘the’

Line 242.  Insert ‘and’ before ‘comorbidities’

Line 243.  Remove ‘the’

Line 245.  Change ‘the longer’ to ‘a longer’

Line 251.  Change ‘patients.’ to ‘patients,’

Line 252.  Change ‘mostly identified five’ to ‘five most commonly identified’

Line 262.  Change ‘anaerobe’ to ‘anaerobes’

Line 277.  Change ‘predictor’ to ‘predictors’

Line 292.  Change ‘vastest’ to ‘greatest’

Line 295.  Insert ‘the’ before ‘30’

Line 315.  Change ‘Consistence’ to ‘Consistent’

Line 333.  Change ‘vaster’ to ‘greater’

Line 386.  Change ‘well-training physician’ to ‘appropriately trained physicians’

Line 396.  Inert a full-stop after ‘results’

Lines 405-406.  Change ‘suggested to routinely perform’ to ‘suggested to be routinely performed’

Page 15.  Change ‘Figure 1. The flowchart’ to ‘Figure 1. Flowchart’

Page 16.  Change ‘Figure 2. The distribution of’ to ‘Figure 2. Distribution of’, ‘indicated a significant difference’ to ‘indicates a significant difference’, ‘Numbers in the bar indicated’ to ‘Numbers in the bar indicate’, and ‘respectively, indicated’ to ‘respectively, indicate’

Page 17.  Change ‘indicated case numbers’ to ‘indicate case numbers’

Author Response

In this manuscript the authors have investigated the incidence of bacteremia and the prognostic effects of prompt administration of antimicrobial therapy (AAT) on nontraumatic out-of-hospital cardiac arrest (OHCA) patients achieving a sustained return of spontaneous circulation (sROSC), compared with non-OHCA patients. The incidence and characteristics of bacteremia were different between the nontraumatic OHCA and non-OHCA patients. It was concluded that performing blood culture sampling and rapid AAT administration as first-aids was essential for nontraumatic OHCA patients following sROSC.

I suggest that this manuscript would be suitable for publication in Antibiotics if subjected to some minor grammatical and typographical changes as follows:

Response: Many thanks for your review and suggestions. We agree your opinions detailing the typo and grammar error. We have reworded these mistakes word-by-word as the following

  1. Line 61.  Change ‘the’ to ‘a’

Response: Corrected. Please refer to line 61 on page 2.

  1. Line 64.  Change ‘as’ to ‘on’

Response: Corrected. Please refer to line 64 on page 2.

  1. Line 71.  Change ‘regarding’ to ‘about’ and ‘characteristic’ to ‘characteristics’

Response: Corrected. Please refer to line 71 on page 2.

  1. Line 88.  Change ‘The case’ to ‘This case’, ‘has been’ to ‘was’ and ‘with the datasets’ to ‘with datasets’

Response: Corrected. Please refer to line 88 on page 3.

  1. Line 90.  Remove ‘, respectively,’ and change ‘The case patient was the’ to ‘Case patients were’

Response: Corrected. Please refer to line 90 on page 3.

  1. Line 91.  Change ‘adult’ to ‘adults’

Response: Corrected. Please refer to line 91 on page 3.

  1. Lines 91-92.  Change ‘The control patient was the non-OHCA patient’ to ‘Control patients were non-OHCA patients’

Response: Corrected. Please refer to line 92 on page 3.

  1. Line 101.  Change ‘record’ to ‘records’

Response: Corrected. Please refer to line 101 on page 3

  1. Line 114.  Remove ‘the’

Response: Corrected. Please refer to line 114 on page 3.

  1. Line 120.  Change ‘well’ to ‘appropriate’

Response: Corrected. Please refer to line 121 on page 3.

  1. Line 121.  Change ‘and the recording’ to ‘and recording’

Response: Corrected. Please refer to line 122 on page 3.

  1. Lines 121-122.  Change ‘by the discussion’ to ‘by discussion’

Response: Corrected. Please refer to line 123 on page 3.

  1. Line 123.  Change ‘consist’ to ‘consisted’

Response: Corrected. Please refer to line 124 on page 3.

  1. Line 135.  Change ‘done’ to ‘taken’

Response: Corrected. Please refer to line 137 on page 4.

  1. Line 195.  Change ‘for the statistical analyses for this study’ to ‘for statistical analysis of this study’

Response: Corrected. Please refer to line 197 on page 5.

  1. Line 212.  Remove ‘who’

Response: Corrected. Please refer to line 214 on page 6.

  1. Line 239.  Change ‘was’ to ‘were’

Response: Corrected. Please refer to line 239 on page 6.

  1. Line 240.  Remove ‘the’

Response: Corrected. Please refer to line 240 on page 6.

  1. Line 242.  Insert ‘and’ before ‘comorbidities’

Response: Corrected. Please refer to line 242 on page 6.

  1. Line 243.  Remove ‘the’

Response: Corrected. Please refer to line 243 on page 6.

  1. Line 245.  Change ‘the longer’ to ‘a longer’

Response: Corrected. Please refer to line 245 on page 6.

  1. Line 251.  Change ‘patients.’ to ‘patients,’

Response: Corrected. Please refer to line 251 on page 6.

  1. Line 252.  Change ‘mostly identified five’ to ‘five most commonly identified’

Response: Corrected. Please refer to line 252 on page 6.

  1. Line 262.  Change ‘anaerobe’ to ‘anaerobes’

Response: Corrected. Please refer to line 262 on page 7.

  1. Line 277.  Change ‘predictor’ to ‘predictors’

Response: Corrected. Please refer to line 277 on page 7.

  1. Line 292.  Change ‘vastest’ to ‘greatest’

Response: Corrected. Please refer to line 292 on page 7.

  1. Line 295.  Insert ‘the’ before ‘30’

Response: Corrected. Please refer to line 295 on page 7.

  1. Line 315.  Change ‘Consistence’ to ‘Consistent’

Response: Corrected. Please refer to line 315 on page 8.

  1. Line 333.  Change ‘vaster’ to ‘greater’

Response: Corrected. Please refer to line 333 on page 8.

  1. Line 386.  Change ‘well-training physician’ to ‘appropriately trained physicians’

Response: Corrected. Please refer to line 386 on page 9.

  1. Line 396.  Inert a full-stop after ‘results’

Response: Corrected. Please refer to line 403 on page 9.

  1. Lines 405-406.  Change ‘suggested to routinely perform’ to ‘suggested to be routinely performed’

Response: Corrected. Please refer to line 411-412 on page 11.

  1. Page 15.  Change ‘Figure 1. The flowchart’ to ‘Figure 1. Flowchart’

Response: Corrected. Please refer to page 16.

  1. Page 16.  Change ‘Figure 2. The distribution of’ to ‘Figure 2. Distribution of’, ‘indicated a significant difference’ to ‘indicates a significant difference’, ‘Numbers in the bar indicated’ to ‘Numbers in the bar indicate’, and ‘respectively, indicated’ to ‘respectively, indicate’

Response: Corrected. Please refer to page 17.

  1. Page 17.  Change ‘indicated case numbers’ to ‘indicate case numbers’

Response: Corrected. Please refer to page 18.

Reviewer 4 Report

Dear authors of the manuscript "blood cultures and appropriate antimicrobial administration after achieving sustained return of spontaneous circulation in adults with nontraumatic out-of-hospital cardiac arrest":

I have read with interest your manuscript, since the issue of ampirical antimicrobial therapy after OHCA is controversial and important.

The introduction and methods are well written.

In the results section, some comment:

P6L212 - "Patients who achieved" or "patients achieving" instead of "Patients who achieving"/

P6L250 - unclear, please rephrase.

Section 1.1, P6L235-242 - You identified clearly the clinical and demographic characteristics of the case patients, that were siginificantly different from the control group. I think it is essential to do some more analysis, propensity score, or multivariate logistic regression, and see if the case patients are different because they are more often  from nursing homes (14.7% vs 5.6%) and more often bed-ridden (35.5% vs 12.6%).

This analysis is mandatory, since it could explain why the case patients have more respiratory tract infections (aspirations maybe?), more group A streptococci and streptococcus pneumoniae, and less gram negative sepsis.

I also have some comments on the discussion section:

P8L324 - 325 - CRP may be elevated because of cardiac arrest, and although it was siginificantly higher than in the control group, clinicaly significance is less obvious when comparing the range of 41-214 to 28-186 (table 1).\

The recommendation in P8L39-342 is not well established in this paper, to my opinion, and should be written with extra caution, especially in the era of MDR pathogens and less new broad spectrum antibiotics.

Also the conclusion, on P10L405 should be rephrased. I do not agree that according to your finding "blood cultures... routinely...".

Author Response

Dear authors of the manuscript "blood cultures and appropriate antimicrobial administration after achieving sustained return of spontaneous circulation in adults with nontraumatic out-of-hospital cardiac arrest": I have read with interest your manuscript, since the issue of empirical antimicrobial therapy after OHCA is controversial and important.

Response: Many thanks for your review.

  1. The introduction and methods are well written.

Response: Greatly thanks for your review.

  1. In the results section, some comment:
  • P6L212 - "Patients who achieved" or "patients achieving" instead of "Patients who achieving"

Response: Thanks for your suggestion. This sentence has been reworded (Line 214, Page 6).

  • P6L250 - unclear, please rephrase.

Response: Thanks for your opinions. This sentence has been rephrased (Line 253-254, Page 6).

  • Section 1.1, P6L235-242 - You identified clearly the clinical and demographic characteristics of the case patients, that were significantly different from the control group. I think it is essential to do some more analysis, propensity score, or multivariate logistic regression, and see if the case patients are different because they are more often from nursing homes (14.7% vs 5.6%) and more often bed-ridden (35.5% vs 12.6%).

This analysis is mandatory, since it could explain why the case patients have more respiratory tract infections (aspirations maybe?), more group A streptococci and streptococcus pneumoniae, and less gram negative sepsis.

Response: Many thanks for your review and suggestions. We agree very much that the difference in patients demographics and clinical manifestations between case and control patients remained significant in our cohort. Herein, we designed to demonstrate the prognostic effects of delayed administration of appropriate antimicrobial therapy by adjusting confounding factors linked to mortality (i.e., our principal outcome), not using the method of propensity-score matching. Taking the covariable of nursing homes as an example, it was one of crucial independent predictors of 30-day mortality, which is necessary to be adjusted, in less critically ill control patients, but it was not regarded as the mortality determinants both in case and critically ill control patients so it is not superfluous to enroll this covariable into the multivariable analysis.

  1. I also have some comments on the discussion section:
  • P8L324 - 325 - CRP may be elevated because of cardiac arrest, and although it was significantly higher than in the control group, clinically significance is less obvious when comparing the range of 41-214 to 28-186 (table 1).

Response: Many thanks for your substantial opinions and suggestions. Quantitative CRP measurement in serum has been proposed as a sensitive and a specific indicator of bloodstream infection (Reference No.23). Notably, this ideal marker reflects not only the presence of sepsis or bacteremia but also its severity (Intensive Care Med 2002; 28:235–243). In the literature, several studies regarding the ability of CRP in predicting outcomes of OHCA patients has been reported, and author believed that CRP may reflect the pre-arrest state of health and thus investigated associations with outcomes (Scientific Reports 2021:11:10279; Int J Clin Pract. 2021;75:e14227). Their finding was consistent with our opinion that bacteremia episodes occurred prior to resuscitation (Line 332-334, Page 8). However, the association of elevated CRP and cardiac arrest has not been established in the literature.  

  • The recommendation in P8L339-342 is not well established in this paper, to my opinion, and should be written with extra caution, especially in the era of MDR pathogens and less new broad spectrum antibiotics.

Response: Many thanks for your substantial opinions and suggestions. To avoid the reader’s misleading, this sentence has rephrased for more conservative statement. Please refer to Line 342-345 on Page 8.

  1. Also the conclusion, on P10L405 should be rephrased. I do not agree that according to your finding "blood cultures... routinely...".

Response: Greatly thanks again for your substantial opinions. Because of a lack of this evidence in our work, this sentence has rephrased for more conservative statement. Please refer to Line 410-413 on Page 11.

This manuscript is a resubmission of an earlier submission. The following is a list of the peer review reports and author responses from that submission.

Round 1

Reviewer 1 Report

I thank the authors for their effort to answer all the Questions I raised. But I do have still Major concerns regarding the publication of this artticle.

The are significant selection bias of the patients and consequently the results are questionable. I maintain my Opionon from the second Review and I do not reccomand this article to be published.

Reviewer 2 Report

The quality of the article has improved significantly with the recommendations suggested by the reviewers. In its current state it could be edited by the journal. 

Reviewer 3 Report

In this revised version authors improved quality of the manuscript.

I suggest acceptance.

Reviewer 4 Report

The current study aims to evaluate the incidence of cardiac arrest secondary to bacteremia. Unfortunately, the study has numerous devastating methodological flaw that compromises the study and its results. Based on prior data, most cases of cardiac arrest are due to cardiac origin and non- cardiac causes account for a small minority. In addition, 30 days OHCA mortality in this study was 91%, similar to previously reported incidence. Thus, it does not appear that bacteremia or antibiotic therapy play a significant role in OHCA mortality.

The authors state on lines 152-154: “reasonable comparators in controlling”. However, based on the study design, I cannot agree that an appropriate control group was selected. The control group include patients that presented to the ED without OHCA. Patients in the control group have less severe disease, are younger, have less comorbidity, more likely to have a UTI or skin infection (Table 1). In addition, only 427 patients (table 2), were reported to be critically ill. The quality of the study would be significantly improved if another control group was selected such as OHCA patients without bacteremia or appropriately match patients based on severity of disease and underlying risk factors.

Finally,  blood cultures were only collected on 55.8% of patients with OHCA. It is likely that blood cultures were not collected on these patients as they did not have symptoms suggestive of infection. If all patients without a blood culture collected were assumed to have a negative blood cultures, rates of OHCA with bacteremia would be 6.7%, closer to the control group bacteremia incidence. Similarly, the bacteria and resistance frequency could be significantly altered by the missing blood cultures.